# Temporal and Spatial Trends in Livestock Manure Discharge and Water Pollution Risk in Chaohu Lake Basin

Fanghui Pan [1,2], Fei Huang [2], Hongguang Zhu [1,*] and Youbao Wang [2,*]

1    School of Mechanical Engineering, Institute of New Rural Development, Tongji University, Shanghai 201804, China; pfh20072008@126.com
2    School of Ecology and Environment, Anhui Normal University, Wuhu 241000, China; hf173810@163.com
*    Correspondence: zhuhg@tongji.edu.cn (H.Z.); wyb74@126.com (Y.W.)

**Abstract:** Assessments of the spatiotemporal distribution of livestock manure and its risk to the watershed are important to prevent water pollution. In this work, the spatiotemporal livestock manure distribution and its risk for the Chaohu lake basin were evaluated based on the excretion coefficient method and ArcGIS technology. In detail, the amounts of livestock manure and its associated pollutants, including chemical oxygen demand (COD), five-day biochemical oxygen demand ($BOD_5$), ammonia ($NH_4^+$-N), total nitrogen (TN), and total phosphorus (TP), were calculated from 2009 to 2019. Then, the diffusion concentrations of COD, $BOD_5$, $NH_4^+$-N, TN, and TP and the water pollution risk index from livestock manure were estimated and predicted for the Chaohu lake basin. The results indicated that the mean amount of livestock manure had reached $1.04 \times 10^7$ t in the Chaohu lake basin in the studied decade. The COD, $BOD_5$, $NH_4^+$-N, TN, and TP from livestock manure in Feixi and Feidong contributed 54.26% and 54.40% of the total in the whole basin. These results demonstrate the potential pollution risk of livestock manure for the Chaohu lake basin. Moreover, the diffusion concentrations of COD, $BOD_5$, $NH_4^+$-N, TN, and TP for the lake basin were from highest to lowest as follows: Feixi > Feidong > Chaohu > Lujiang > Wuwei > Shucheng > Hefei. The water pollution risk index was more than 20 in Feixi and Feidong, indicating that these areas were heavily affected by local livestock manure. The water pollution risk index will be approximately 18 for the Chaohu lake basin in 2030, implying that the Chaohu lake watershed will suffer moderate pollution from animal manure. These results provide scientific support for policymakers to enhance manure utilization efficiency and control livestock manure loss, causing water eutrophication in Chaohu lake basin or other similar watersheds.

**Keywords:** Chaohu lake basin; animal manure; diffusion concentration; water pollution; risk assessment

## 1. Introduction

Sizeable and intensive livestock industries have been emerging throughout the world to meet the increasing population's demand for animal-based foods [1,2], resulting in massive manure production [3,4]. However, inadequate animal manure collection and storage has led to the extremely low utilization efficiency of manure in cropland and contributed to high losses of manure nutrients to water [1,5,6]. Chinese data showed that animals excreted $2.3 \times 10^7$ t N nationally; however, only one-third of the total amount was recycled to cropland in 2010 [5], with losses via leaching, runoff, and erosion amounting to $3.4 \times 10^6$ t of N [7]. In addition, the direct discharge of manure to water bodies or landfills contributed 78% to the total manure P losses and 61% to the total manure K losses [7]. A large proportion of the rivers, lakes, and coastal waters in China suffer from severe eutrophication, and approximately 46% of the rivers in China have been classified as harmful to direct human contact; a major contributor of nutrients to watercourses, which causes eutrophication, is industrialized animal production systems [5]. Hence, it is

essential to predict the spatiotemporal distribution of livestock manure and assess its risk for watersheds in order to prevent its loss causing water eutrophication.

Currently, the spatiotemporal variation tendency of livestock manure has been studied to prevent regional environmental pollution [8–10]. Yan et al. [10] estimated the spatiotemporal distribution of livestock and poultry farms in the Shangjie region, China, using ArcGIS technology, and the loss rate of livestock manure was considered. Jia et al. [9] estimated manure production and its geographical distribution in China and strengthened the potential for manure reuse using scientific composting for sustainable and environmental aims. Gallengo et al. [8] offered GIS and multicriteria tools to assess livestock farms with animal waste for sectorial, environmental, and social risks, particularly the relationship between the high density of livestock in an area and the levels of eutrophication of surface water and groundwater.

In China, lakes and reservoirs are important strategic water resources, but the frequent eutrophication and water blooms of lakes and reservoirs have caused harm to aquatic ecosystem health and drinking water safety [11,12]. Chaohu lake has been listed as one of the "Three Rivers and Lakes" under the national water pollution prevention and control [13]. It is one of the largest freshwater lakes in China [13–15]. With the rapid development of the social economy in the catchments, it has become one of the most hypereutrophic lakes since the 1980s [15–17]. There was a wide range of blue-green algae bloom outbreaks in Chaohu lake in 2018 [12,18,19]. The eutrophication and blue-green algae bloom in the western part of Chaohu lake were more serious than those in the eastern part due to the main pollution occurring in the west of the Chaohu lake basin [20,21]. Currently, most research has concentrated on the harm to lakes caused by hazardous materials for lakes based on the hydrodynamic model [11], evaluation of transient storage, and nutrient retention, such as N and P, in lakes [12,22]. Nevertheless, the risk effect of animal manure on lakes or reservoirs has rarely been investigated in China.

Therefore, in order to effectively prevent animal manure loss from including water eutrophication, the spatiotemporal distribution of animal manure and its risk effect on water pollution were investigated and evaluated via a case study of the Chaohu lake basin. The goals of this work were to (1) quantify the amount of animal manure, including pig dung equivalent and its associated pollutants (COD, $BOD_5$, $NH_{4+}$-N, TN, and TP) in different years (from 2009 to 2019), and obtain the spatial distribution at the county scale of Chaohu lake basin; and (2) evaluate the risk for Chaohu lake basin via perspective analysis of the diffuse concentration of pollutants and the water pollution risk index of animal manure utilizing ArcGIS technology. The study provides a scientific basis for decision makers to reasonably restructure animal farms and optimize their spatial layout to improve the sustainable development of animal husbandry and protect the water environment.

## 2. Materials and Methods

### 2.1. Study Area and Data Sources

Chaohu lake basin (30°56′30″ N to 32°16′18″ N, 116°25′42″ E to 118°20′23″ E) is situated at the center of Anhui province, eastern China, between the Yangtze and Huai rivers. The Chaohu lake basin area is 13,486 km$^2$ [13,15,23]. A humid subtropical monsoon climate prevails in the area, with a mean annual temperature of 16 °C and a mean annual rainfall of 1215 mm [24]. The Chaohu lake is long from east to west with a length of 55 km, short from north to south with a width of 22 km, high in the west, and low in the east and middle [25,26]. The lake is made up of nine tributaries: Nanfei river, Shiwuli river, Pai river, Hangbu river, Zhegao river, Shuangqiao river, Zhao river, Baishitian river, and Yuxi river [20]. The Chaohu lake basin covers all counties and cities of Feidong, Feixi, Lujiang, Shucheng, Hefei city, Chaohu, and Wuwei (Figure 1).

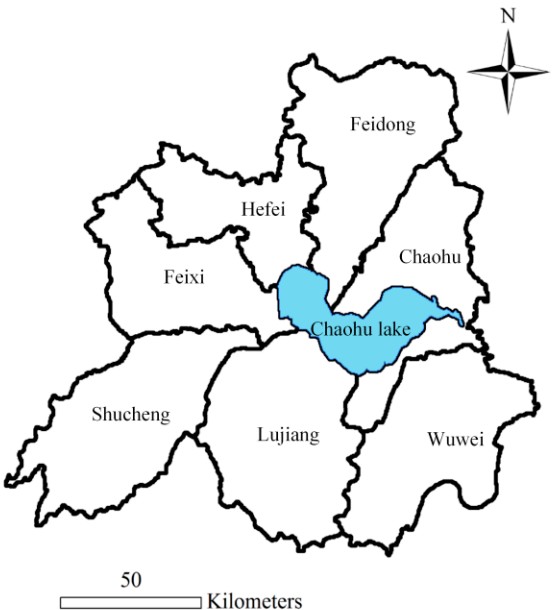

**Figure 1.** Chaohu lake basin and its counties and cities.

The numbers of animal husbandry and total water resources from 2009 to 2019 are derived from the statistical yearbook on the local bureau of statistics websites (http://tjj.ah.gov.cn; http://tjj.hefei.gov.cn; http://tjj.wuhu.gov.cn; http://tjj.luan.gov.cn) accessed on 10 December 2020 and the relevant literature from Web of Science (http://apps.webofknowledge.com/) accessed on 11 December 2020 and China National Knowledge Infrastructure (CNKI) (http://www.cnki.net/) accessed on 12 December 2020. The numbers of animal husbandry are shown in Table S1, and the gross amount of water resources is shown in Table S2.

## 2.2. Computing the Quantity of Animal Manure in Chaohu Lake Basin

### 2.2.1. Pig Dung Equivalent

The pig equivalent was calculated to simplify the calculation of the number of livestock and poultry units, using the conversion coefficient between the pigs and other livestock and poultry to represent all livestock and poultry units [10]. Based on the excretion coefficient of animals and the feeding cycle (Table S3) [27–29], the amount of animal manure is computed using Equation (1):

$$Q_i = N_i \times k_i \times T_i , \tag{1}$$

where $Q_i$ is the number of tons of animal manure annually, $t \cdot a^{-1}$; $N_i$ is the number of livestock and poultry, head or capita; $k_i$ is the excretion coefficient, which is defined as the daily discharge of animal manure by weight, $kg \cdot d^{-1}$ [9]; $T_i$ is the raising cycle of animals, days; i denotes the animal type: cow, pig, sheep, or poultry.

In accordance with Equation (1) and the conversion coefficient of the pig dung equivalent of animal manure (Table S4), the pig dung equivalent was obtained, as shown in Equation (2):

$$Q'_i = Q_i \times \mu_i, \tag{2}$$

where $Q'_i$ is the pig dung equivalent, $t \cdot a^{-1}$; $\mu_i$ is the conversion coefficient of the pig dung equivalent.

### 2.2.2. Pollutant Content from Animal Manure

According to the concentration coefficients of the pollutants regarding COD, $BOD_5$, $NH_4^+$-N, TN, and TP from animal manure (Table S5), presented by Song et al. [29] and

Wang et al. [30], the respective amounts of COD, BOD$_5$, NH$_4^+$-N, TN, and TP from animal manure are calculated using Equation (3):

$$M_j = Q_i \times \delta_j \, , \tag{3}$$

where $M_j$ is the amount of pollutant j in animal manure per year, t·a$^{-1}$; $\delta_j$ is the concentration coefficient of pollutant j in animal manure, kg·t$^{-1}$; j refers to COD, BOD$_5$, NH$_4^+$-N, TN, or TP in animal manure.

### 2.3. Diffusion Concentration and Water Pollution Risk from Animal Manure

### 2.3.1. Diffusion Concentration of Animal Manure

Losses of animal manure to surface water occur via leaching, runoff, and erosion [31,32]. The loss rate of animal manure is inconsistent due to the fact that the number of livestock farms, manure management, local climates, and other factors are diverse in different regions, and the available data show that the loss rate is between 20% and 40% in China [29]. Therefore, the loss rate of animal manure is defined as the percentage of livestock manure loss due to leaching, runoff, and erosion. The loss rate of animal manure is roughly 30% in this study because of the Chaohu lake basin's affluent water resources [25]. On the basis of the pollutant content of animal manure (Equation (3)) and the water resources (Table S2), the diffusion concentrations of the pollutants in animal manure are computed in Equation (4):

$$C_j = \frac{M_j \times l}{V}, \tag{4}$$

where $C_j$ is the diffusion concentration of pollutants j, from among COD, BOD$_5$, NH$_4^+$-N, TN, and TP, from animal manure, respectively, mg·L$^{-1}$; V is the gross water resource in Chaohu lake basin, m$^3$; l is the loss rate of animal manure.

### 2.3.2. Water Pollution Risk Index from Animal Manure

Based on the water quality standard level (Table S4) (MEEPRC, 2002) [33] and Equation (4), the water pollution risk index from animal manure is depicted in Equation (5) as follows:

$$I = \sum_{j=1}^{5} \frac{C_j}{C_{0j}}, \tag{5}$$

where I is the water pollution risk index from animal manure; $C_{0j}$ is the concentration of water quality (Table S6), mg·L$^{-1}$.

According to Equation (5), the grade of water pollution risk is ulteriorly classified as five levels, as seen in Table 1.

**Table 1.** Water pollution risk index and its level [30,34].

| I | $\leq 5$ | $5 < I \leq 10$ | $10 < I \leq 15$ | $15 < I \leq 20$ | $> 20$ |
|---|---|---|---|---|---|
| level | I | II | III | IV | V |
| extent of pollution | none | slight | moderate | severe | very severe |

### 2.3.3. Risk Prediction for Water Pollution from Animal Manure

According to the method of forecasting livestock excrement offered by Wang et al. (2021) [30], the risk of animal manure for water pollution is forecasted for 2030, as shown in Equation (6).

$$I_{\text{future-value}} = I_{\text{mean-value}} \left\{ 1 + \left[ \left( \frac{I_{\text{mean-value}}}{I_{\text{initial-value}}} \right)^{\frac{1}{n_{\text{final-year}} - n_{\text{initial-year}}}} - 1 \right] \right\}^{\frac{1}{n_{\text{future-year}} - n_{\text{final-year}}}}, \tag{6}$$

where $I_{\text{future-year}}$ is the water pollution risk value from animal manure in 2030; $I_{\text{mean-value}}$ is the mean water pollution risk value from animal manure from 2009 to 2019; $I_{\text{initial-value}}$ is the water pollution risk value from animal manure in 2009; $I_{\text{final-value}}$ is the water pollution risk value from animal manure in 2019; $n_{\text{initial-year}}$ represents 2009; $n_{\text{final-year}}$ represents 2019; $n_{\text{future-year}}$ represents 2030.

The spatiotemporal distribution of the risk of animal manure for water systems is further assayed using the Kriging Interpolating method. The method is based on spatial statistics, giving different weights to each sample point to minimize the estimation error according to the relationship between the spatial position of the sample points and the correlation degree [30,35]. The spatial distribution of the risk magnitude of animal manure for the Chaohu lake basin is displayed using ArcGIS 10.2 software. Chaohu lake basin is divided into 237 grids. Each grid cell is 9.98 km × 10.23 km from west to east and from north to south (Figure 2) and represents the water pollution risk index of animal manure (I), according to each county or city location of the Chaohu lake basin. Microsoft Excel 2010 software was used to calculate and analyze all the original statistical data of livestock manure in the Chaohu lake basin. Origin 9.0 software was used to depict the statistical results in graphical form.

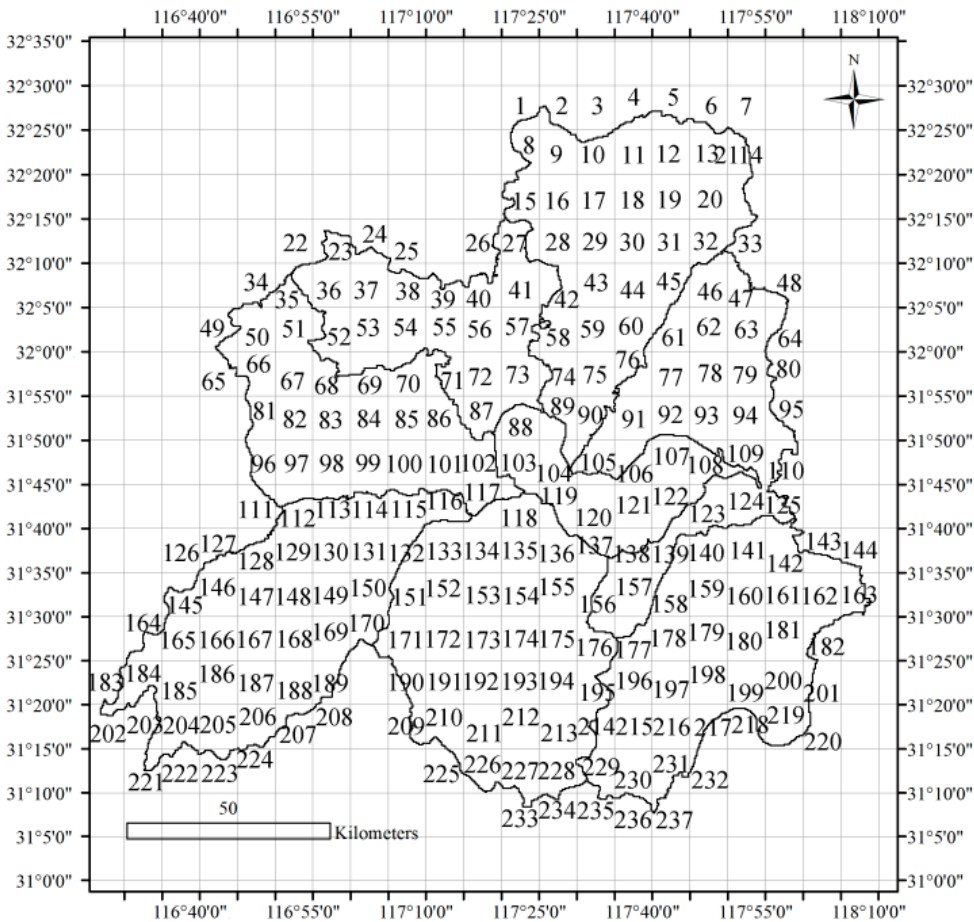

**Figure 2.** Grid distribution points in Chaohu lake basin.

## 3. Results

### 3.1. The Quantity of Animal Manure in Chaohu Lake Basin

3.1.1. Pig Dung Equivalent Production

The pig dung equivalent in the Chaohu lake basin from 2009 to 2019 is shown in Figure 3. The pig dung equivalent showed a light fluctuating increase from 2009 to 2017 and an obvious decrease from 2018 to 2019. The decrease in the pig dung equivalent

was related to the decline in the number of cows, pigs, and poultry in Feidong and Feixi counties (Table S1). This decrease in the number of pigs resulted from African swine fever's introduction and spread in 2018 in China leading to the rapid death of almost all the infected animals [36]. The maximum value of pig dung equivalent reached $1.14 \times 10^7$ t in 2015; however, the pig dung equivalent was at $8.36 \times 10^6$ t and $8.32 \times 10^6$ t in 2018 and 2019, respectively. The sum of the pig dung equivalent in Feixi and Feidong was more than 50% of the total pig dung equivalent in the entire basin (Figure 3). This was followed by Shucheng and Wuwei, where the total pig dung equivalent was 20% of that in the entire basin; then, Hefei, Lujiang, and Chaohu accounted for 18–20% of that in the entire basin. These results indicate that the animal manure from Feixi, Feidong, Shucheng, and Wuwei easily poses a threat to Chaohu lake because of the massive production of pig dung equivalent in those watersheds.

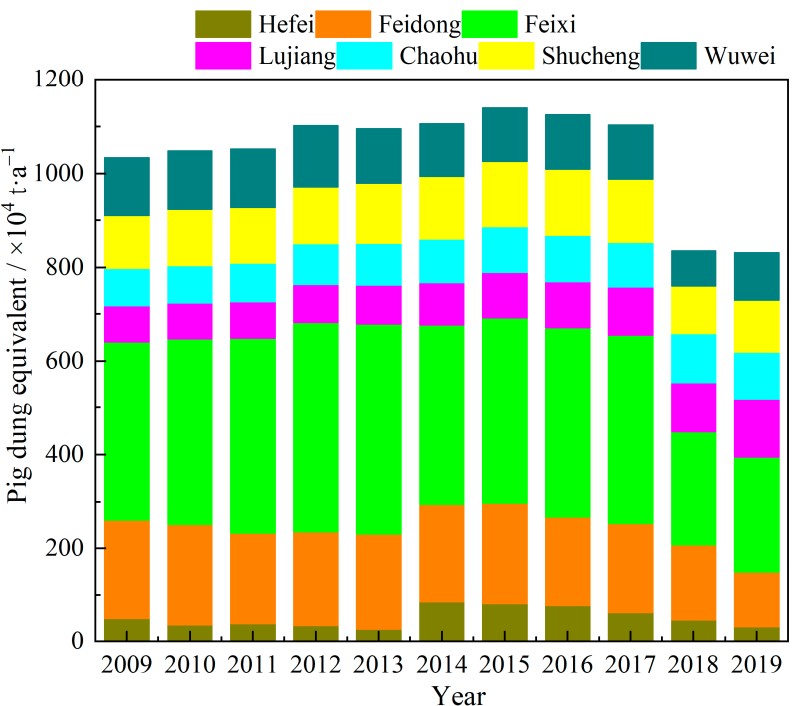

**Figure 3.** The amount of animal manure in Chaohu lake basin from 2009 to 2019.

### 3.1.2. The Amount of Pollutants in Animal Manure

Figure 4 displays the variety of pollutants from animal manure in the Chaohu lake basin in the studied decade (from 2009 to 2019). The animal-manure-associated COD, $BOD_5$, $NH_4^+$-N, TN, and TP contents showed a slight fluctuating increasing trend from 2009 to 2017 and a decreasing tendency in 2018 and 2019 (Figure 4a–e). The COD content of animal manure was in the range of $2.49 \times 10^5$ t and $2.74 \times 10^5$ t, and the $BOD_5$ content of animal manure ranged from $2.19 \times 10^5$ t to $2.37 \times 10^5$ t from 2009 to 2017 in the entire basin (Figure 4a,b). The COD and $BOD_5$ contents were approximately $2.02 \times 10^5$ t and $1.78 \times 10^5$ t in 2018 and 2019, respectively (Figure 4a,b). The maximum contents of COD and $BOD_5$ reached $2.74 \times 10^5$ t and $2.41 \times 10^5$ t in 2015, respectively. Both the sum of the COD content and the total $BOD_5$ content from animal manure in Feixi and Feidong occupied between 45% and 60% of the total for the entire basin. This was followed by Shucheng and Wuwei, with 21% and 25% of the total, respectively. The total COD content and the $BOD_5$ derived from animal manure in Hefei, Lujiang, and Chaohu accounted for approximately 18% and 30% of the total, respectively (Figure 4a,b).

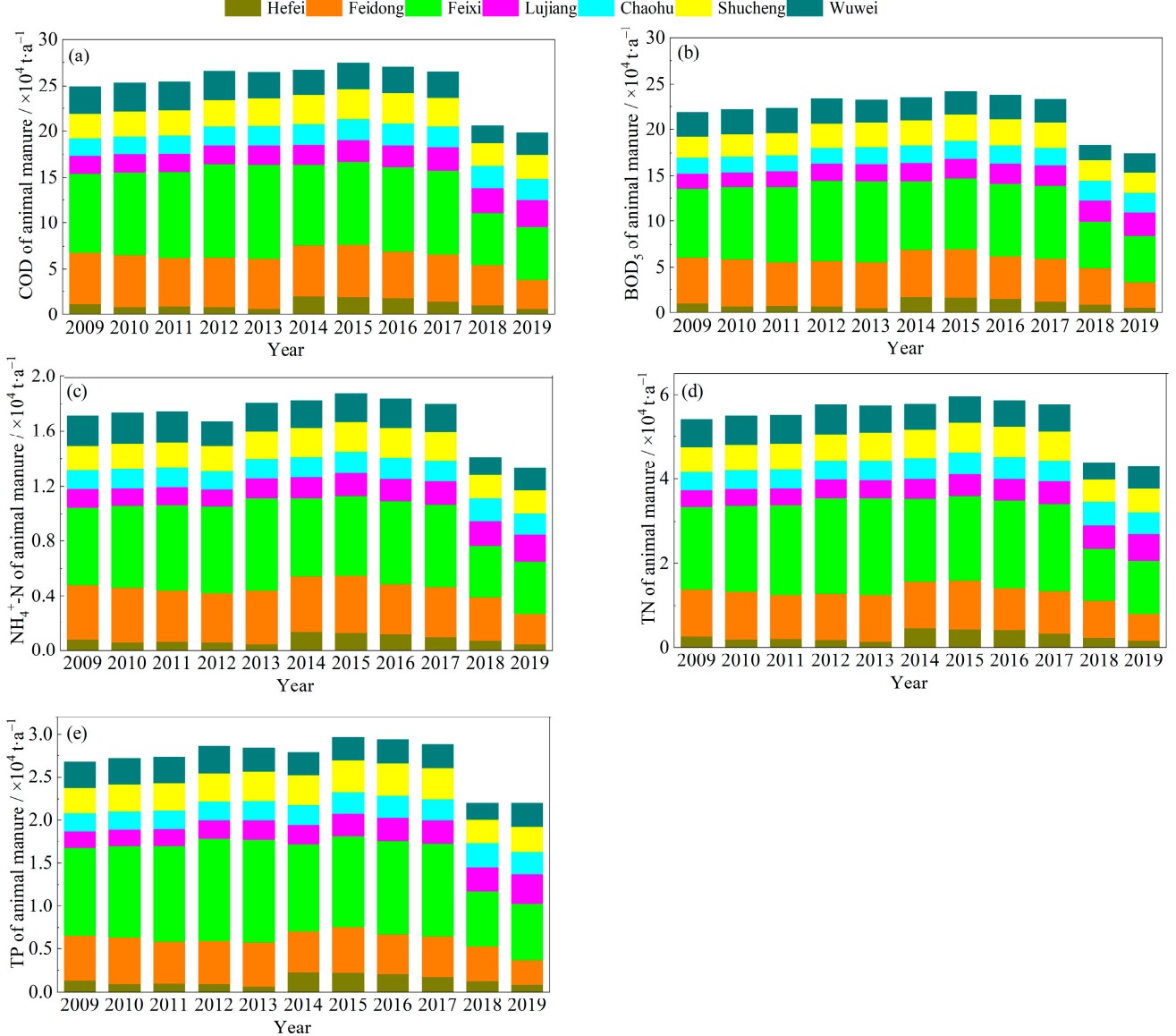

**Figure 4.** The amount of pollutants from animal manure in the Chaohu lake basin from 2009 to 2019. (**a**) COD of animal manure, (**b**) BOD$_5$ of animal manure, (**c**) NH$_4^+$-N of animal manure, (**d**) TN of animal manure, and (**e**) TP of animal manure.

The NH$_4^+$-N and TN contents from animal manure were in the range of 1.67 × 10$^4$ t to 1.87 × 10$^4$ t and 5.41 × 10$^4$ t to 5.95 × 10$^4$ t, from 2009 to 2017, respectively (Figure 4c,d). The NH$_4^+$-N and TN contents from animal manure in the Chaohu lake basin were roughly 1.37 × 10$^4$ t and 4.35 × 10$^4$ t between 2018 and 2019, respectively. In 2015, the NH$_4^+$-N and TN contents of animal manure showed peak values for the entire basin, reaching 1.87 × 10$^4$ t and 5.95 × 10$^4$ t, respectively. The sum of the NH$_4^+$-N and TN contents from animal manure in Feixi and Feidong was 54% of the total in the basin. Shucheng and Wuwei provided approximately 23%, and Hefei, Lujiang, and Chaohu accounted for about 23% of the total.

The average TP content from animal manure reached 2.71 × 10$^4$ t in the entire lake basin in recent years (Figure 4e). In 2015, the TP content reached its maximum value of 2.97 × 10$^4$ t. The TP content of animal manure in Feixi and Feidong provided approximately 54% of the total. Shucheng and Wuwei contributed about 22%, and Hefei, Lujiang, and Chaohu accounted for approximately 23% of the total in the entire basin.

These results indicate that the animal manure contained not only a high content of organic substances (COD of 24 kg/t pig dung and $BOD_5$ of 21 kg/t pig dung) but also abundant nitrogen ($NH_4^+$-N of 1.6 kg/t pig dung and TN of 5.2 kg/t pig dung) and phosphorus (TP of 2.6 kg/t pig dung), which are likely to induce eutrophication in Chaohu lake if animal manure is lost to surface water via runoff. In particular, the loss of nitrogen and phosphorus from livestock manure causes lake eutrophication [24]. As Chaohu lake is one of the five freshwater lakes in China, and the basin is a typical basin where water problems are concentrated and complex [13], pollutant runoff from local livestock manure is only one aspect affecting eutrophication in the basin.

### 3.2. Influence of Animal Manure on the Water in Chaohu Lake Basin

3.2.1. Pollutant Diffusion Concentration from Animal Manure

The fluctuating change in the COD, $BOD_5$, $NH_4^+$-N, TN, and TP diffusion concentrations from animal manure in the Chaohu lake basin over the ten years is shown in Figure 5. The mean diffusion concentration of COD and $BOD_5$ reached 110.08 mg·L$^{-1}$ and 96.93 mg·L$^{-1}$ in the basin from 2009 to 2019, respectively (Figure 5a,b). The diffusion concentrations of COD and $BOD_5$ separately reached maximum values of 147.36 mg·L$^{-1}$ and 129.87 mg·L$^{-1}$ in 2019 and minimum values of 62.09 mg·L$^{-1}$ and 54.63 mg·L$^{-1}$ in 2012. In addition, the average diffusion concentration of COD and $BOD_5$ for the entire basin during the ten years was sequenced as follows: Feixi > Feidong > Chaohu > Lujiang > Wuwei > Shucheng > Hefei.

In the Chaohu lake basin, the mean diffusion concentrations of $NH_4^+$-N and TN reached 7.47 mg·L$^{-1}$ and 23.83 mg·L$^{-1}$ during the decade (Figure 5c,d). The maximal diffusion concentrations of $NH_4^+$-N and TN were 10.00 mg·L$^{-1}$ and 31.98 mg·L$^{-1}$ in 2019, respectively, while the lowest diffusion concentrations of $NH_4^+$-N and TN were 4.25 mg·L$^{-1}$ and 13.45 mg·L$^{-1}$, respectively. It was reported that the TN concentration in Chaohu lake reached 2.17 mg·L$^{-1}$ in 2018, an increase of 2% compared with 2012 [21]. The TN diffusion concentration from animal manure was far beyond that in Chaohu lake. The result indicated that the TN from animal manure could lead to the deterioration of the water quality in Chaohu lake. Additionally, the diffusion concentration of $NH_4^+$-N and TN in the entire basin from 2009 to 2019 was as follows: Feixi > Feidong > Chaohu > Lujiang > Wuwei > Shucheng > Hefei.

The mean diffusion concentration of TP was 11.85 mg·L$^{-1}$ in the basin from 2009 to 2019 (Figure 5e). In 2019, the diffusion concentration of TP reached its highest value of 16.12 mg·L$^{-1}$. The lowest diffusion concentration of TP was 6.69 mg·L$^{-1}$ in 2016. Zhang et al.'s [21] study indicated that the TP concentration was 0.13 mg·L$^{-1}$ in Chaohu lake in 2018, higher by 17% than in 2012. The diffusion concentration of TP from animal manure in the basin was almost 90 times more than 0.13 mg·L$^{-1}$. This result revealed that the TP loss from animal manure was extremely serious in the lake basin, potentially causing water pollution in Chaohu lake. In addition, the diffusion concentration of TP in the entire basin in these ten years was in the following order: Feixi > Feidong > Chaohu > Lujiang > Wuwei > Shucheng > Hefei.

3.2.2. Water Pollution Risk Index from Animal Manure

Figure 6 shows the influence of the water pollution risk caused by animal manure in the Chaohu lake basin in this decade. In 2009, the water pollution risk index was between 3.69 and 64.38, and a relatively high-risk index was shown in Feixi and Feidong (Figure 6a). In 2014, the water pollution risk index from animal manure was between 6.12 and 62.75 in the lake basin; therein, Feixi and Feidong still showed a relatively high water pollution risk index (Figure 6b). In 2019, the variation in the water pollution risk index due to animal manure was between 4.02 and 73.06; simultaneously, the water pollution risk index in Feixi and Feidong was still high (Figure 6c). Based on Equation (6), the water pollution risk caused by animal manure in the entire basin in the future was predicted. The water pollution risk index due to animal manure is likely to be between 5.5 and 48.3 in the basin

in 2030, and the high water pollution risk index in Feixi and Feidong is likely to be 47.7 and 28.4, respectively (Figure 6d).

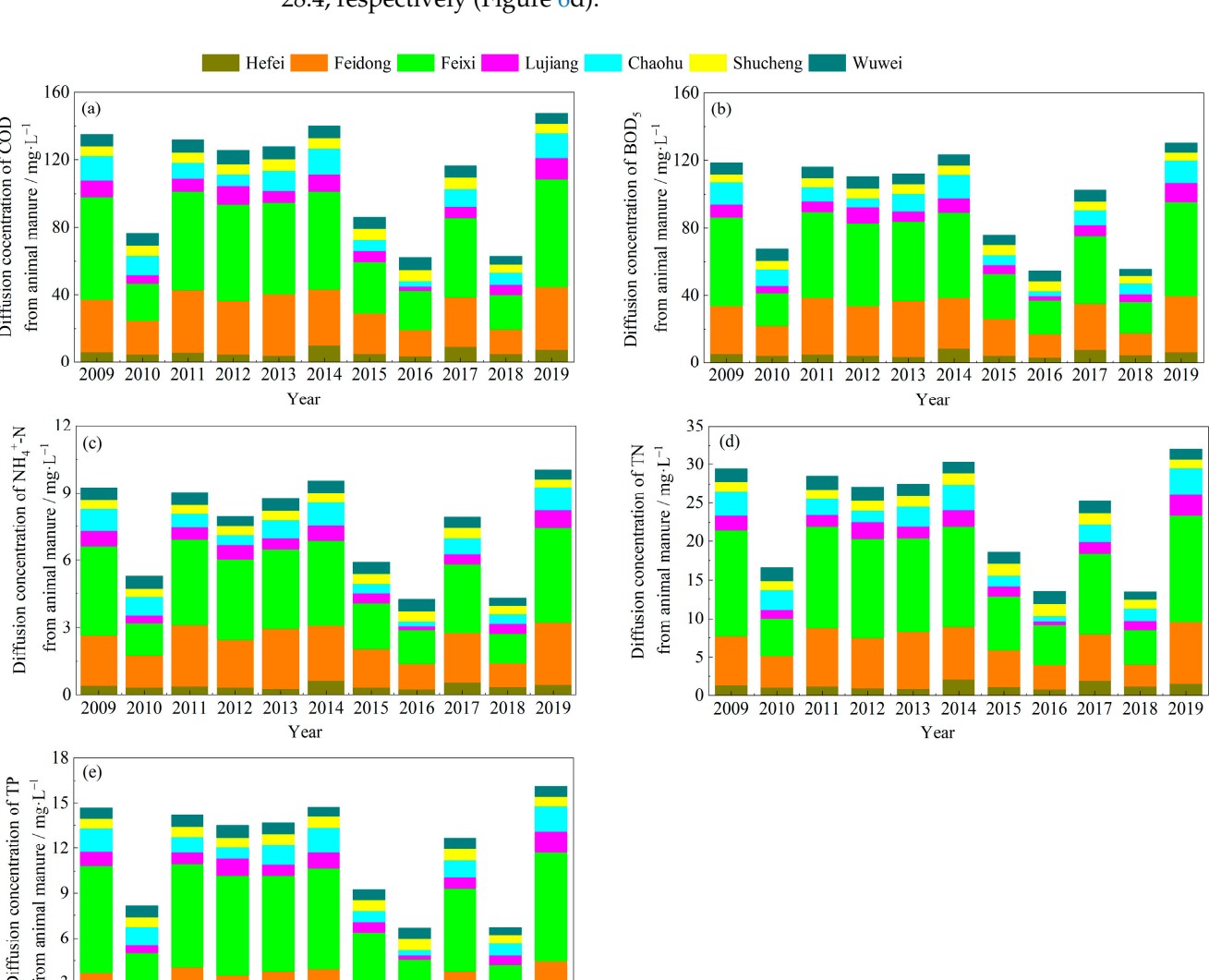

**Figure 5.** Pollutant (COD, BOD$_5$, NH$_4^+$-N, TN, and TP) diffusion concentration from animal manure in the Chaohu lake basin. (**a**) the diffusion concentration of COD, (**b**) the diffusion conenctration of BOD$_5$, (**c**) the diffusion concentration of NH$_4^+$-N, (**d**) the diffusion concentration of TN, and (**e**) the diffusion concentration of TP.

Based on the magnitude of the water pollution risk (Table 1), in 2009, the watershed suffered very severe pollution (V) in Feixi and Feidong due to animal manure, moderate pollution (III) in Chaohu, and slight pollution (II) in Wuwei, Lujiang, Hefei, and Shucheng. In 2014, the watershed showed very severe pollution (V) in Feixi and Feidong, moderate pollution (III) in Chaohu and Hefei, and slight pollution (II) in Lujiang, Shucheng, and Wuwei. In 2019, the watershed in Feixi and Feidong suffered very severe pollution (V), followed by Chaohu with severe pollution (IV), Lujiang with moderate pollution (III), and finally, Hefei, Wuwei, and Shucheng with slight pollution (II). In 2030, the watershed in Feixi and Feidong is likely to be subject to very severe pollution (V) due to animal manure, followed by Chaohu, with severe animal manure pollution (IV). In Lujiang and Hefei, animal manure will cause moderate pollution (III) in the watershed, with Shucheng and Wuwei subject to slight pollution (II).

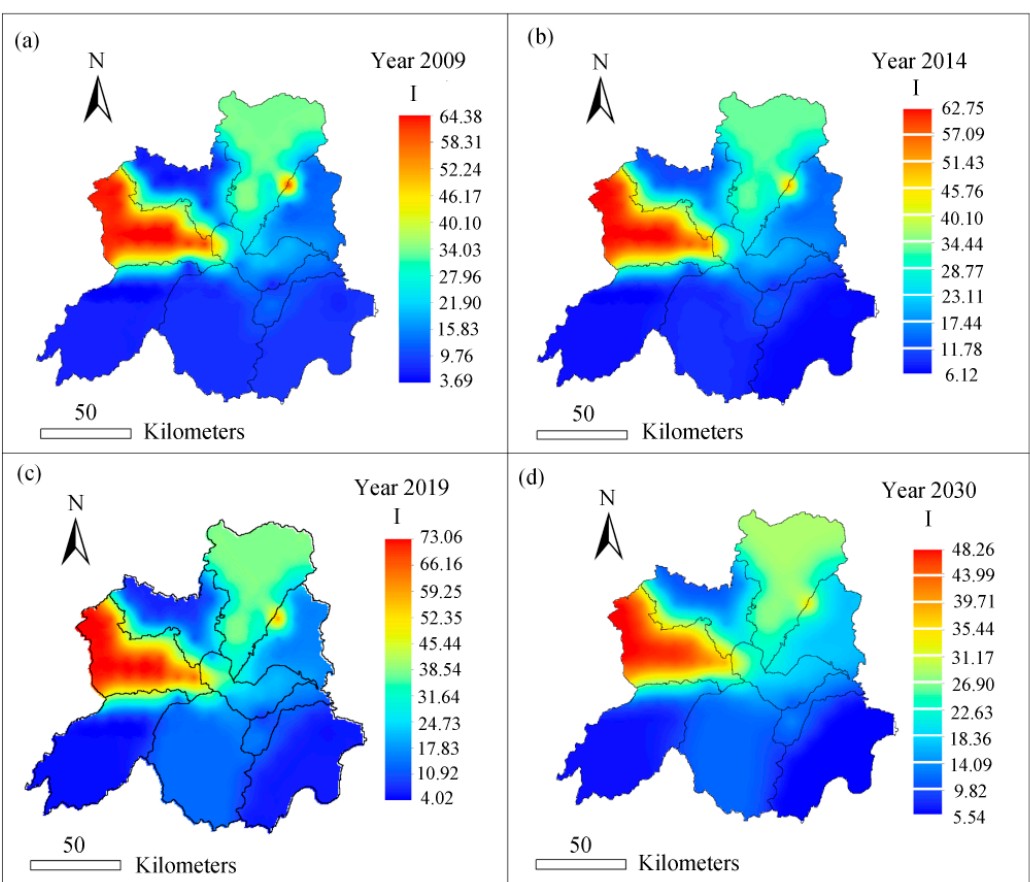

**Figure 6.** Water pollution risk index of animal manure in Chaohu lake basin. (**a**) the water pollution risk index (I) in 2009, (**b**) the water pollution risk index (I) in 2014, (**c**) the water pollution risk index (I) in 2019, and (**d**) the predicted water pollution risk index (I) in 2030.

## 4. Discussion

On account of the high water pollution risk of animal manure in the Chaohu lake basin, animal manure poses a threat to the whole basin [37,38]. Previous studies [20,22,39] indicated that eutrophication was worse in the western part of Chaohu lake than in the eastern part due to the nitrogen and phosphorus discharge of anthropogenic activities, including the built livestock farms. The major pollution sources, such as animal manure, are mainly distributed in the western basin (Figure 6). However, there are animal manure losses via leaching and runoff [5,23]. It was reported that a large proportion of manure N from pig production is lost via direct discharge into water bodies or landfilling [7].

Table 2 shows the influence of every 1% decrease in the loss rate for COD, BOD$_5$, NH$_4^+$-N, TN, and TP from animal manure in the Chaohu lake basin in 2019. Table 3 shows the influence of the reduction in the loss rate on the different water pollution risk levels (I) in the basin in 2019. Table 4 displays the influence of the reduction in the loss rate from animal manure for different water pollution risk levels (I) in the basin in 2030. The results showed that the water pollution induced by the pollutants COD, BOD$_5$, NH$_4^+$-N, TN, and TP from livestock manure would be alleviated annually if the loss rate decreased by 1%. The water risk index would decline at the same time. The risk index (I) of animal manure for Chaohu lake would be less than 5 if the loss rate of animal manure was reduced by 10.65%. These results revealed that animal manure losses have a strong influence on water pollution. Adequate animal manure collection and storage in animal farms should be prioritized to prevent manure losses.

**Table 2.** Influence of the loss rate with a 1% reduction in the pollutants from animal manure in Chaohu lake basin in 2019.

| Loss Rate | Chaohu Lake Basin | Loss of Animal Manure for Every 1% Reduction in the Loss Rate in the Basin in 2019 (t·a$^{-1}$) | | | | |
|---|---|---|---|---|---|---|
| | | COD | BOD$_5$ | NH$_4^+$-N | TN | TP |
| 1% | Hefei | 71.44 | 62.06 | 4.62 | 15.85 | 8.42 |
| | Feidong | 305.34 | 273.40 | 22.30 | 64.44 | 29.28 |
| | Feixi | 577.66 | 506.48 | 38.15 | 125.84 | 65.61 |
| | Lujiang | 296.01 | 259.78 | 19.52 | 64.20 | 33.56 |
| | Chaohu | 231.25 | 203.78 | 15.66 | 51.78 | 26.17 |
| | Shucheng | 260.86 | 228.69 | 17.18 | 56.57 | 29.61 |
| | Wuwei | 235.82 | 205.06 | 15.44 | 52.36 | 27.43 |
| | Whole basin | 1978.38 | 1739.25 | 132.87 | 431.05 | 220.08 |

**Table 3.** Effects of the decrease in the loss rate of the animal manure pollutants for the underwater pollution risk level in Chaohu lake basin in 2019.

| Chaohu Lake Basin | I Value in 2019 | I Value for Every 1% Reduction in the Loss Rate | Decrease Value of the Loss Rate (%) | | | |
|---|---|---|---|---|---|---|
| | | | I = 5 | I = 10 | I = 15 | I = 20 |
| Hefei | 8.45 | 0.28 | 12.25 | - | - | - |
| Feidong | 38.63 | 1.29 | 26.12 | 22.24 | 18.35 | 14.47 |
| Feixi | 71.68 | 2.39 | 27.91 | 25.81 | 23.72 | 21.63 |
| Lujiang | 13.88 | 0.46 | 19.19 | 8.39 | - | - |
| Chaohu | 16.90 | 0.56 | 21.12 | 12.25 | 3.37 | - |
| Shucheng | 6.14 | 0.20 | 5.56 | - | - | - |
| Wuwei | 6.72 | 0.22 | 7.69 | - | - | - |
| Whole basin | 23.20 | 0.77 | 23.53 | 17.07 | 10.60 | 4.14 |

**Table 4.** Effects of the decrease in the loss rate of the animal manure pollutants for the underwater pollution risk level in Chaohu lake basin in 2030.

| Chaohu Lake Basin | I Value in 2030 | I Value for Every 1% Reduction in the Loss Rate | Decrease Value of Loss Rate (%) | | | |
|---|---|---|---|---|---|---|
| | | | I = 5 | I = 10 | I = 15 | I = 20 |
| Hefei | 6.08 | 0.28 | 3.83 | - | - | - |
| Feidong | 24.21 | 1.29 | 14.92 | 11.03 | 7.15 | 3.27 |
| Feixi | 34.80 | 2.39 | 12.47 | 10.38 | 8.29 | 6.19 |
| Lujiang | 5.93 | 0.46 | 2.00 | - | - | - |
| Chaohu | 7.05 | 0.56 | 3.64 | - | - | - |
| Shucheng | 7.74 | 0.20 | 13.40 | - | - | - |
| Wuwei | 6.84 | 0.22 | 8.20 | - | - | - |
| Whole basin | 13.23 | 0.77 | 10.65 | 4.18 | - | - |

Environmental pollution (water, soil, and air) from livestock and poultry farms could be prevented and controlled by incorporating indicators (diffusion concentration, loss rate, and risk index) for the environmental pollution potential of livestock manure to evaluate the suitable spatial distribution of livestock and poultry farms [8–10]. The results of this study provide scientific and technical support for livestock and poultry management and decision making.

## 5. Conclusions

The spatiotemporal distribution of livestock manure discharge and its water pollution risk was estimated in the Chaohu lake basin based on the excretion coefficient method and ArcGIS technology. The amount of livestock manure was $1.04 \times 10^7$ t in the Chaohu lake basin from 2009 to 2019, with an increasing and then decreasing trend. The pollutants COD,

BOD$_5$, NH$_4^+$-N, TN, and TP from livestock manure in Feixi and Feidong accounted for 54.26% and 54.40% of the total in the basin. The results show the potential risk for the entire basin. Further, the diffusion concentrations of COD, BOD$_5$, NH$_4^+$-N, TN, and TP in the entire lake basin from 2009 to 2019 were in the following order: Feixi > Feidong > Chaohu > Lujiang > Wuwei > Shucheng > Hefei. The water pollution risk index in Feixi and Feidong was higher than 20, indicating that these regions suffered serious pollution from animal manure. In 2030, the water pollution risk index is predicted to be approximately 18 for the Chaohu lake basin, revealing that the Chaohu lake watershed will have moderate pollution from animal manure. These results provide a scientific basis for policymakers to undertake some effective measures, including coupling planting and breeding, changing manure into organic fertilizer, and reducing the manure loss rate to improve the manure utilization efficiency and prevent manure's direct discharge into water bodies in the Chaohu lake basin.

**Supplementary Materials:** The following supporting information can be downloaded at: https://www.mdpi.com/article/10.3390/su16062396/s1, References [29,30,33,40] are cited in the Supplementary Materials.

**Author Contributions:** Conceptualization, H.Z. and Y.W.; Methodology, Y.W.; Software, F.P., F.H. and H.Z.; Validation, F.P. and H.Z.; Formal analysis, F.P., H.Z. and Y.W.; Investigation, F.P. and F.H.; Resources, H.Z. and Y.W.; Data curation, F.P., F.H. and H.Z.; Writing—original draft, F.P.; Writing—review & editing, Y.W.; Visualization, H.Z.; Project administration, H.Z. and Y.W.; Funding acquisition, H.Z. All authors have read and agreed to the published version of the manuscript.

**Funding:** This work was supported by the Special Project of Municipal-school Cooperation in Science and Technology [grant number SXHZ202206] and Shanghai "Super Postdoc" incentive program [grant number 2023789].

**Institutional Review Board Statement:** Not applicable.

**Informed Consent Statement:** Not applicable.

**Data Availability Statement:** No new data were created or analyzed in this study. Data sharing is not applicable to this article.

**Conflicts of Interest:** The authors declare no conflicts of interest.

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
