# Peer review of "Temporal and Spatial Trends in Livestock Manure Discharge and Water Pollution Risk in Chaohu Lake Basin"

_sustainability, doi:10.3390/su16062396_

Round 1

Reviewer 1 Report

Comments and Suggestions for Authors

Abstract

The abstract is not clear to me, I didn’t get the grasp of what the authors were trying to say. I will suggest you rewrite this section for clarity and easy understanding.

Introduction

The introduction generally was hard to understand. The arguments for the topic were not well thought out. For instance, the authors did not mention how the manure came about, the risk it poses to the water environment as specified on the title. There are so many disjointed sentences, and the paragraphs were not connected. A reader would like to see a sequence of argument that is easy to understand and comprehend. This section is also too shallow, now detail arguments while you think this topic is what publishing, what are the implications of your study? The authors need to look at the grammar again. Further, the authors did not follow the referencing style of the journal, please adapt.  See the attached document for detailed comments.

Materials and methods

I find it hard to understand what the authors did. The study area description was not clear, I didn’t see anything about the geography, climate and land use activities in the studied area. The land use(s) is the main driver of pollutants the authors are claimed to have looked at in this study. Further, most of the statistical tools or equations the authors claimed to have used did not align with the parameters that they defined. For instance, in lines 91-95, equation 1 formula was different from the defined parameters. I saw Table 2 but no Table 1, why? I am wondering if the study design was well thought out.  Please see the attached document for more comments.

Results

The is section was not well presented. I could not understand what the authors were presenting. Please interpret the results in clear terms. I kept seeing a value was achieved, did you achieve values or recorded values in the course of your research? I would like to see a better way of results interpretation.

Discussion

The authors concentrated on results interpretation here instead of discussing their findings.

Conclusion

I didn’t look at this section, as I there are much to be done by the authors before a conclusion can be reached.

Comments on the Quality of English Language

Extensive English edit is required.

Reviewer 2 Report

Comments and Suggestions for Authors

Dear Authors, 

The article submitted for review concerns an attempt to show how the immediate industrial environment can affect the water quality of Chaohu Lake (巢湖). The authors focused on the effects of pollution produced by livestock and poultry farms located close to this water reservoir. The topic deserves attention and publication. However, to be of scientific value and potentially have an impact on regional environmental policy, it is necessary to complete many of the topics raised and sort out what the authors actually want to say. 

1. The abstract is written in a chaotic manner. The authors have lost sight of the purpose of this article or have not yet fully thought of what this publication is supposed to serve. This is missing throughout the manuscript. It is difficult to follow the subsequent paragraphs because they seem to be random and there is no common main idea. A scientific article is not exactly about reporting on the analyses performed.

2. There are abbreviations of thought which are not necessarily readable by anyone other than the authors. In addition, there are some unfortunate phrases such as livestock farms produce excessive amounts of excrement ( line10-11) - animal excrement is neither excessive nor too little - it is what it is. Here, after all, the point is to make it clear that meat producers do not collect animal manure properly.

3. There is a lack of explanation of what is meant by abbreviations such as COD, BOD5, etc. How were the values of the different parameter concentrations described in the manuscript calculated? Little is known about how the authors obtained reliable pollution data. Was it just an analysis of some local database or was it their own measurements?

4. There is a lack of explanation of what the authors mean by using the term diffusion concentration of contaminants. 

5. The suggestions for solving the problem given in the summary are far too general. The impression is given that the topic is not well thought through and therefore it is difficult to show how it could be solved. 

6. Do the values given for the equivalent of pig faeces represent a high or low value? In addition to this, what do the percentages described in the line detailing the types of pollutants flowing into the lake actually show? What do they really refer to and why are they so important to the authors.

7. Although the lake area has been divided into individual sections, there is a definite lack of background to the analysis, i.e., showing what is farmed at a particular part of the 巢湖 and in what quantity (if only approximately). This could combine information about how many animals live in the immediate area and whether the amount of manure they produce cause the increase in lake pollution in that particular part. 

8. What does it really mean for the authors to improve the sustainability of animal husbandry and water quality in 巢湖 (Lines 65-67)? At least three different topics have been mixed in one sentence, making it completely unclear what this could be about. Is it really about the need to make livestock farming more sustainable or is it perhaps an attempt to show a lack of adherence to safe storage of livestock manure? 

9. There is no information on why 巢湖 is so important to the local ecosystem. Maybe it supplies fresh water to the surrounding homes? Perhaps there are some unique animals, plants, etc. which could extinct by the mismanagement of faeces? 

10. There is a missing paragraph showing what animal faeces pollution leads to for human health (zoonoses) and for the aquatic ecosystem or local wildlife. This information is needed to show how important this topic is and why the authors took the time to address it. 

11. The summary is insufficient. There is a lack of analysis and a real presentation of what can be done about this problem. Moreover, the last sentence is almost the same as the last sentence in the abstract (even grammatical errors have been copied).

Sincerely Yours, 

Reviewer

Comments on the Quality of English Language

Dear Authors, 

The manuscript submitted for review is written mainly in an imprecise and ill-considered manner, and this is the major reason why the English is also of poor quality. Therefore, I do not point out specific places or words where errors occur. Giving up too long sentences and replacing them with shorter ones will certainly help.

Sincerely Yours, 

Reviewer

Reviewer 3 Report

Comments and Suggestions for Authors

Dear authors,

Thank you for the possibility to review to paper titled "Temporal and Spatial Evolution of Livestock Manure Discharge and Its Pollution Risk to Water Environment in Chaohu Lake Basin" that presents a comprehensive study on the environmental impact of animal manure from intensive livestock farms in the Chaohu Lake basin. It evaluates the risk of water pollution through a decade-long analysis (2009-2019) of manure discharge, assessing the quantities of pig dung equivalent, pollutant concentrations, and their diffusion into the lake. The study utilizes statistical principles and ArcGIS technology to predict future risks and suggests sustainable practices like converting manure to organic fertilizer to mitigate pollution. The findings underscore the critical need for improved manure management to protect water quality in the region. Here are some recommendations for improvement that might be considered:

-    -    Please present in introduction chapter what you will present in the rest of the paper;

-     -   Please in the methodology part include a clearer explanation of the Kriging Interpolating method and its appropriateness for the data set.

-      - Provide access to raw data for transparency or include a supplementary material section where data can be examined by others for validation purposes.

- -       Discuss the policy implications of the study more thoroughly and suggest specific policy changes or interventions based on the findings.

-      -  Clearly outline areas where further research is needed, particularly in exploring long-term trends beyond the prediction models used in the study.

Comments on the Quality of English Language
    • Conduct a thorough review for typographical errors, grammatical mistakes, and ensure consistency in terminology throughout the paper.

Reviewer 4 Report

Comments and Suggestions for Authors

An improved and refined version of the title, can be:

Temporal and Spatial Trends in Livestock Manure Discharge and Water Pollution Risk in Chaohu Lake Basin

 1.      The abstract lacks clarity in explaining the specific methodologies used to estimate the diffusion concentration of pollutants from animal manure and predict water pollution risk. Also, the abstract could benefit from clearer organization and structure to improve readability and from a clearer statement of the study's objectives and research questions.

 1.Introduction: Providing a clearer rationale for why Chaohu Lake was chosen as the focus of the study would enhance the coherence of the introduction.

2.      The introduction highlights the potential benefits of the study for restructuring animal farms and improving water quality in Chaohu Lake area, but it could provide more specific examples or implications to demonstrate the practical relevance of the research findings.

 1.      The Materials and Methods section provides detailed equations and calculations, but it lacks clarity in explaining some of the terms and parameters used. For example, the excrement coefficient of animals and the conversion coefficient of pig dung equivalent are mentioned without sufficient explanation or reference to their sources.

2.      The description of the grid distribution points in the Chaohu Lake basin is somewhat unclear and could be improved with more specific details on how the grids were delineated and what factors were considered in their selection. Clarity on these points would enhance the reproducibility of the spatial analysis.

3.      While the section mentions the use of ArcGIS software for spatial analysis, it does not provide details on how this software was used or the specific functions employed. Providing more information on the GIS methods used would enhance the technical rigor of the spatial analysis.

Line 94: amount

 1.      The Results section provides detailed information on various parameters and trends, and the presentation could be improved for clarity. Some sentences are lengthy and contain multiple ideas, making it challenging to follow the flow of information. Breaking down complex sentences into smaller, segments would enhance readability and comprehension.

2.      In some instances, section presents findings without providing sufficient context or explanation. For example, the text mentions fluctuations in pollutant concentrations without discussing potential factors driving these fluctuations, such as changes in agricultural practices or environmental regulations. Providing additional context and interpretation would strengthen the scientific rigor of the study and help readers interpret the results more effectively.

3.      Lines 255-267: keep the same font dimension all over the text

 1.In the "Discussion" section is addresses the significance of reducing animal manure loss rates, however the contextualization within broader environmental management strategies is not underline. Integrating discussions on the feasibility, cost-effectiveness, and potential trade-offs associated with implementing such measures would enrich the analysis and offer practical insights for decision-makers.

2. Potential uncertainties or limitations associated with the presented findings are not mentioned. Addressing factors such as variability in environmental conditions, data quality, and the assumptions underlying the analysis would strengthen the credibility and robustness of the conclusions.

3. The discussion does not address potential uncertainties or limitations associated with the presented findings. Given the complexity of environmental systems and the inherent variability in pollutant dynamics, acknowledging and discussing uncertainties would enhance the credibility and reliability of the conclusions.

 1.While the conclusions mention several measures to improve manure utilization efficiency and prevent direct discharge into croplands and water bodies, they lack specificity regarding implementation strategies. Providing concrete recommendations and policy suggestions would enhance the practical relevance of the conclusions for stakeholders and policymakers.

 2.The prediction of moderate pollution in the watershed of Chaohu lake in 2030 is mentioned but not fully explained. It would be beneficial to provide context regarding the factors driving this prediction and potential implications for water quality management strategies.

 The provided references seem to cover a wide range of topics related to animal manure management, water pollution, and environmental assessments in the Chaohu lake basin and similar regions.

See also the comments in the text of the manuscript. 

Round 2

Reviewer 1 Report

Comments and Suggestions for Authors

Dear authors, 

Many thanks for the revised version of your manuscript. This version is an improvement of the earlier version. However, you still need to improved the flow of argument in the introduction section. For instance, the paragraph in lines 49-59 does not flow with the preceding paragraph. This and many other arguments need to be reworked for easy understanding.

Further, I mentioned in my earlier comments, that you did not follow the referencing style of mdpi, but here you did not attend to that comment. Kindly do that in your resubmission.

In line 207 you mentioned in the studied decade, what do you mean by that?

In line 233, change content to concentration.

In line 246: "basin over the ten years is" should be changed to "basin over the ten years" should be changed to basin for the period of study.

Overall, recheck the grammar for clarity.

Comments on the Quality of English Language

Minor revision is needed.

Reviewer 2 Report

Comments and Suggestions for Authors

Dear Authors, 

The manuscript submitted for review presents a different quality. The manuscript is coherent and reads well. The authors have re-edited the text, which has helped to show what the purpose of this work was. The responses to the reviews are very precise, but for some reason they are not all included in the current version of the manuscript. It can be accepted that some of them will be, as it is, a private response to the doubts and ambiguities that arose when reading the first version of the manuscript. However, I think it is important to briefly describe why this Chaohu Lake is so important (as what is in the manuscript does not answer this question at all). I think these three statements from the review response should be added: Chaohu Lake has been listed as one of the 'Three Rivers and Lakes' under National Water Pollution Prevention and Control. It is one of the largest freshwater lakes in China. But it has become one of the most hypereutrophic lakes since the 1980s with the rapid development of the social economy in the catchments.'

Moreover, add at the very beginning of the manuscript, as soon as the explanations for abbreviations like COD, BOD5, etc. start to be used.

Sincerely Yours, 

Reviewer
